# Structure of scavenger receptor SCARF1 and its interaction with lipoproteins

Yuanyuan Wang[1,2,3†], Fan Xu[1†], Guangyi Li[4†], Chen Cheng[1], Bowen Yu[5], Ze Zhang[1], Dandan Kong[1], Fabao Chen[1], Yali Liu[1], Zhen Fang[1], Longxing Cao[6], Yang Yu[4], Yijun Gu[4], Yongning He[1,2,3,7,8]*

[1]State Key Laboratory of Systems Medicine for Cancer, Shanghai Cancer Institute, Renji Hospital, Shanghai Jiao Tong University School of Medicine, Shanghai, China; [2]Shanghai Institute of Biochemistry and Cell Biology, Center for Excellence in Molecular Cell Science, Chinese Academy of Sciences, Shanghai, China; [3]University of Chinese Academy of Sciences, Beijing, China; [4]National Facility for Protein Science in Shanghai, Shanghai Advanced Research Institute, Chinese Academy of Sciences, Shanghai, China; [5]Department of Immunology, School of Basic Medical Sciences, Weifang Medical University, Weifang, China; [6]School of Life Science, Westlake University, Hangzhou, China; [7]Shanghai Key Laboratory for Cancer Systems Regulation and Clinical Translation, Shanghai, China; [8]Department of Biliary-Pancreatic Surgery, Renji Hospital, Shanghai Jiao Tong University School of Medicine, Shanghai, China

*For correspondence: heyn@shsmu.edu.cn

[†]These authors contributed equally to this work

## eLife Assessment

SCARF1 is a scavenger membrane-bound receptor that binds modified versions of lipoproteins and has a significant role in maintaining lipid homeostasis. This **useful** study reports the crystal structure of SCARF1 and identifies putative binding sites for modified lipoproteins. Supported by a **convincing** set of experimental approaches, this study advances our knowledge of how scavenger receptors clear modified lipoproteins to maintain lipid homeostasis.

**Abstract** SCARF1 (scavenger receptor class F member 1, SREC-1 or SR-F1) is a type I transmembrane protein that recognizes multiple endogenous and exogenous ligands such as modified low-density lipoproteins (LDLs) and is important for maintaining homeostasis and immunity. But the structural information and the mechanisms of ligand recognition of SCARF1 are largely unavailable. Here, we solve the crystal structures of the N-terminal fragments of human SCARF1, which show that SCARF1 forms homodimers and its epidermal growth factor (EGF)-like domains adopt a long-curved conformation. Then, we examine the interactions of SCARF1 with lipoproteins and are able to identify a region on SCARF1 for recognizing modified LDLs. The mutagenesis data show that the positively charged residues in the region are crucial for the interaction of SCARF1 with modified LDLs, which is confirmed by making chimeric molecules of SCARF1 and SCARF2. In addition, teichoic acids, a cell wall polymer expressed on the surface of gram-positive bacteria, are able to inhibit the interactions of modified LDLs with SCARF1, suggesting the ligand binding sites of SCARF1 might be shared for some of its scavenging targets. Overall, these results provide mechanistic insights into SCARF1 and its interactions with the ligands, which are important for understanding its physiological roles in homeostasis and the related diseases.

## Introduction

Scavenger receptor (SR) was first discovered in late 1970s during the studies regarding the accumulation of low-density lipoprotein (LDL) in macrophages in atherosclerotic plaques of patients who lack LDL receptors (*Goldstein et al., 1979*; *Brown and Goldstein, 1979*). Up to date, SR family includes a large number of cell surface proteins that can be classified into more than 10 classes (class A-L) based on the structural similarities (*PrabhuDas et al., 2017*). SRs bind a wide range of endogenous and exogenous ligands including modified lipoproteins, damaged or apoptotic cells, and pathogenic microorganisms (*Brown and Goldstein, 1983*; *Dunne et al., 1994*; *Krieger et al., 1993*; *Means et al., 2009*; *Cheng et al., 2019*) and play important roles in maintaining homeostasis, host defense, and immunity, and have been linked to diseases such as cardiovascular diseases, Alzheimer's disease, and cancer (*Huang et al., 2019*; *Zani et al., 2015*; *Yu et al., 2015*; *Patten et al., 2022*; *Wilkinson and El Khoury, 2012*).

Scavenger receptor class F (SR-F) has two known members, SCARF1 and SCARF2 (*PrabhuDas et al., 2017*: *Figure 1A*). SCARF1 was identified in cDNA libraries from human umbilical vein endothelial cells as a receptor for modified LDLs, thus also named SREC-1 (scavenger receptors expressed in endothelial cells) (*Adachi et al., 1997*). It is expressed on the surface of endothelial cells, macrophages, and dendritic cells and distributed in organs such as heart, liver, kidney, and spleen (*Tamura et al., 2004*). SCARF1 recognizes modified LDLs, including acetylated LDL (AcLDL), oxidized LDL (OxLDL), and carbamylated LDL (*Adachi et al., 1997*; *Apostolov et al., 2009*; *Sano et al., 2012*). Previous studies have shown that the SCARF1-mediated degradation of AcLDL accounts for 60% of the amounts of AcLDL degraded by the pathway independent of scavenger receptor class A (SR-A), suggesting that SCARF1 may play a key role in the development of atherosclerosis in concert with SR-A in some situations (*Tamura et al., 2004*). SCARF1 expressed on antigen-presenting cells such as dendritic cells can recognize and internalize heat shock protein-bound antigens and activate adaptive immune responses (*Gong et al., 2009*; *Murshid et al., 2010*; *Murshid et al., 2014*). It may also cooperate with the Toll-like receptors to mediate the cytokine production (*Means et al., 2009*; *Jeannin et al., 2005*; *Beauvillain et al., 2010*). SCARF1-knockout mice can develop symptoms similar to systemic lupus erythematosus disease and lead to accumulation of apoptotic cells in the immune organs, suggesting that it is involved in the removal of apoptotic cells and maintaining homeostasis (*Ramirez-Ortiz et al., 2013*). SCARF1 may also associate with the extravasation of leukocytes from circulation into inflamed tissues during injury or infection, thus having a role in the inflammatory changes in vessel walls and the initiation of atherosclerosis (*Ishii et al., 2002*; *Patten et al., 2017*). Recent data also show that SCARF1 is down-regulated in hepatocellular carcinoma and loss of SCARF1 is associated with poorly differentiated tumors (*Patten et al., 2020*). SCARF2 has 35% sequence identity and similar organ distribution with SCARF1 (*Ishii et al., 2002*). Genetic analysis suggests that SCARF2 is linked to a rare disease called van den Ende-Gupta syndrome (*Anastasio et al., 2010*; *Migliavacca et al., 2014*), which may provide clues for the physiological roles of this molecule.

Previous reports have shown that SCARF1 can bind a number of endogenous ligands other than modified LDLs, including heat shock proteins (*Murshid et al., 2010*; *Murshid et al., 2014*), calreticulin (*Berwin et al., 2004*), Ecrg4 (*Moriguchi et al., 2018*), Tamm-Horsfall protein (*Pfistershammer et al., 2008*), and apoptotic cells (*Ramirez-Ortiz et al., 2013*; *Jorge et al., 2022*), and mediate ligand internalization and transport (*Berwin et al., 2004*; *Narazaki et al., 2008*). It can also bind bacterial, viral, and fungal antigens (*Means et al., 2009*; *Beauvillain et al., 2010*; *Rechner et al., 2007*; *Schade and Weidenmaier, 2016*), but the mechanisms for having such diverse ligand binding properties are unclear. By contrast, SCARF2 shows no binding activity with modified LDLs (*Ishii et al., 2002*; *Wicker-Planquart et al., 2021*), but recent data suggest SCARF2 may share ligands such as complement C1q and calreticulin with SCARF1 (*Wicker-Planquart et al., 2021*). In addition, MEGF10 (multiple epidermal growth factor [EGF]-like domains-10), which might be another member of SR-F family, is a mammalian ortholog of Ced-1 and a receptor of amyloid-β in the brain (*Wilkinson and El Khoury, 2012*; *Nagase et al., 2001*; *Suzuki and Nakayama, 2007*), suggesting that SR-F family members may have rather wide ligand binding specificities.

SCARF1 (MW, 86 kD) is a type I transmembrane protein that has a short signal peptide followed by a long extracellular region, a transmembrane helix, and a large cytoplasmic portion (*Adachi et al., 1997*; *Figure 1A*). Its ectodomain has three glycosylation sites and contains multiple EGF-like domains, which

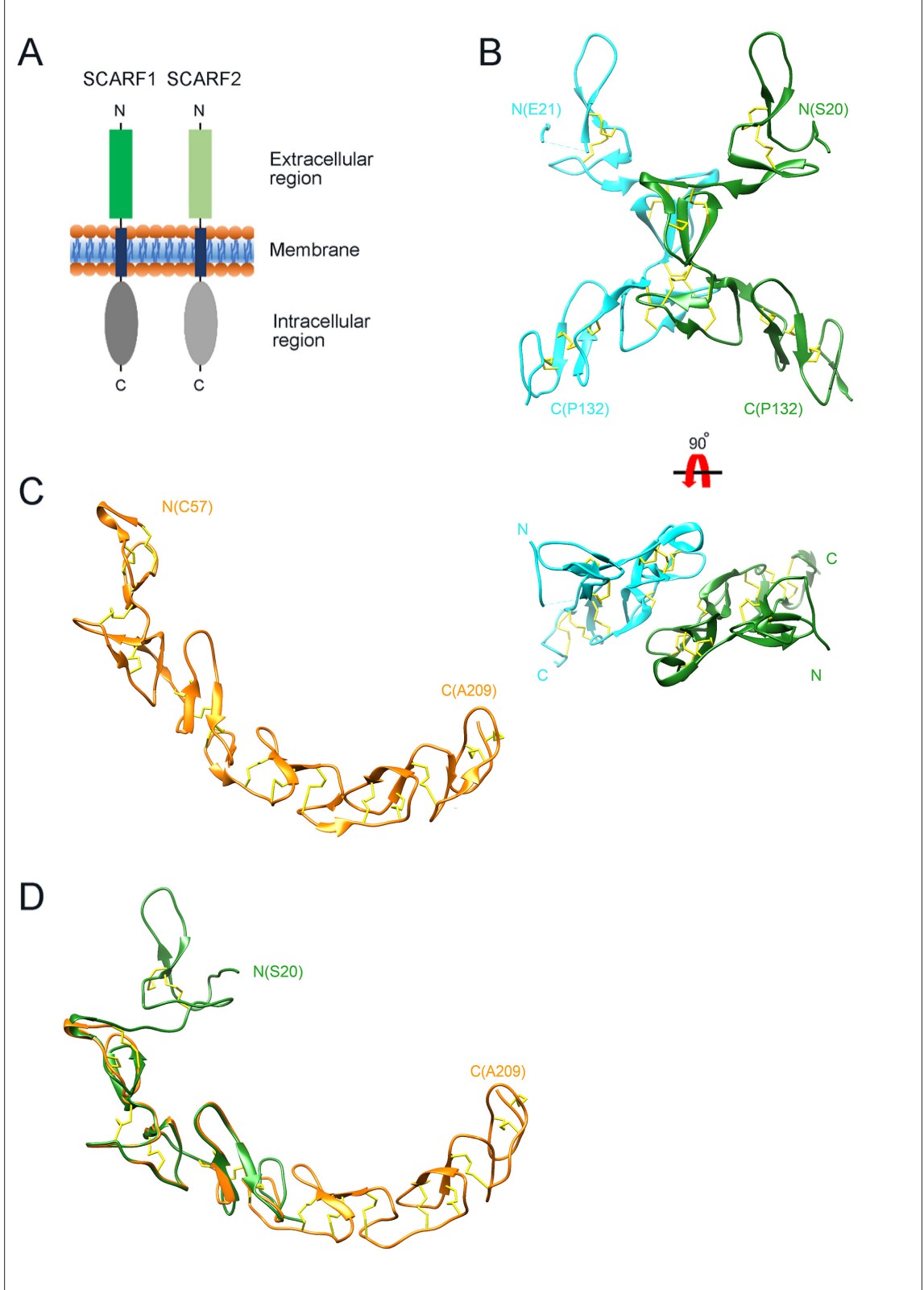

**Figure 1.** Crystal structure of the N-terminal fragments of SCARF1. (**A**) A schematic model of human SCARF1 and SCARF2. (**B**) Ribbon diagrams of a homodimer of an N-terminal fragment (f1, 20–132aa; two monomers are shown in cyan and green, respectively) of SCARF1. (**C**) A ribbon diagram of an N-terminal fragment (f2, 57–209aa, gold) of SCARF1. (**D**) Structure of the N-terminal fragment of SCARF1 (20–209aa) by superimposing the crystal structures of f1 (green) and f2 (gold). Disulfide bonds are shown in yellow (**B–D**).

*Figure 1 continued on next page*

*Figure 1 continued*

The online version of this article includes the following figure supplement(s) for figure 1:

**Figure supplement 1.** The size exclusion chromatography (SEC) profiles of the SCARF1 fragments for crystallization.

**Figure supplement 2.** Models of the ectodomains of SCARF1 and SCARF2 by AlphaFold.

usually has a two-stranded β-sheet followed by a loop region and three conserved disulfide bonds (*Adachi et al., 1997*; *Sano et al., 2012*; *Bork et al., 1996*). It has been shown that modified LDLs are the ligands for several SRs, including SR-A members, scavenger receptor class B type I (SR-BI), lectin-like oxidized LDL receptor (LOX-1), CD36, etc. (*Gillotte-Taylor et al., 2001*; *Endemann et al., 1993*; *Sawamura et al., 1997*; *Cheng et al., 2021*). For the SR-A members, including SCARA1 (CD204, SR-A1), MARCO, and SCARA5, they bind modified LDLs in a Ca$^{2+}$-dependent manner through the scavenger receptor cysteine-rich domains of the receptors (*Cheng et al., 2021*). In the case of LOX-1, both positively charged and non-charged hydrophilic residues might be involved in lipoprotein recognition (*Shi et al., 2001*; *Ohki et al., 2005*). And for CD36, positively charged residues are identified to be critical for its binding with OxLDL (*Kar et al., 2008*). Since SCARF1 has different structural features with other known lipoprotein receptors, how it recognizes modified lipoproteins remains unclear.

Here, we determined the crystal structures of the N-terminal fragments of SCARF1 and characterized the interaction of SCARF1 with modified LDLs by biochemical and mutagenesis studies, thus providing mechanistic insights into the recognition of lipoproteins by this receptor.

## Results

### Crystal structures of the N-terminal fragments of SCARF1

To determine the structure of the extracellular region of SCARF1, the intact and several truncation fragments of SCARF1 ectodomain were expressed in insect cells and purified for crystallization screening. Among them, two fragments (f1, 20–132aa and f2, 20–221aa) (*Figure 1—figure supplement 1*) were crystallized and the crystals diffracted to 2.2 Å and 2.6 Å, respectively (*Supplementary file 1*, *Supplementary file 2*). The initial phasing for f1 crystal was done by SAD using Pt derivatives and the structure was refined to 2.2 Å with a native dataset after molecular replacement. The crystal of f1 fragment belongs to space group P212121 with two molecules per asymmetric unit (*Figure 1B*). The f1 crystal structure contains a number of loops and two stranded β-sheets stabilized by hydrogen bonds and disulfide bonds, which is consistent with the typical feature of EGF-like domains, and it adopts a bow-like conformation with the middle part (~55–102aa) protruding outward (*Figure 1B*). The electron density of the N-terminal end (~21–62aa) of one molecule in an asymmetric unit is relatively weak and some residues are missing, probably due to the flexibility of this region.

The structure of f2 fragment was solved by molecular replacement using the structure of f1 fragment combined with the EGF-like fragments predicted by AlphaFold as phasing models and refined to 2.6 Å resolution. The crystal belongs to space group P4122 with one molecule per asymmetric unit (*Figure 1C*). In the f2 crystal structure, the N-terminal region of SCARF1 (residue 20–56aa) and the C-terminal region (210–221aa) are largely missing, suggesting these regions are quite flexible, consistent with low electron density of the N-terminal end of the f1 crystal. Rest of the f2 fragment adopts a long-curved conformation with multiple EGF-like domains arranged in tandem (*Figure 1C*). Superposition of the two crystal structures reveals the structure of the N-terminal fragment of SCARF1 (20–209aa) (*Figure 1D*), which has similar structural features with the AlphaFold prediction of the molecule (*Figure 1—figure supplement 2A*). However, the crystal structures show a much larger curvature and local differences with the AlphaFold model (*Figure 1—figure supplement 2A*, *Figure 2—figure supplement 1C*), which is in agreement with a recent report regarding the comparison between the protein structures in PDB and AlphaFold models (*Terwilliger et al., 2024*).

### SCARF1 forms homodimers

In the crystal of f1 fragment, two molecules in an asymmetric unit are related by a twofold noncrystallographic symmetry axis. Interestingly, in the crystal of f2 fragment, although an asymmetric unit has only one molecule, the dimer related by the crystallographic twofold symmetry can be well superimposed with the dimer found in the asymmetric unit of f1 crystals (*Figure 2A*), suggesting that

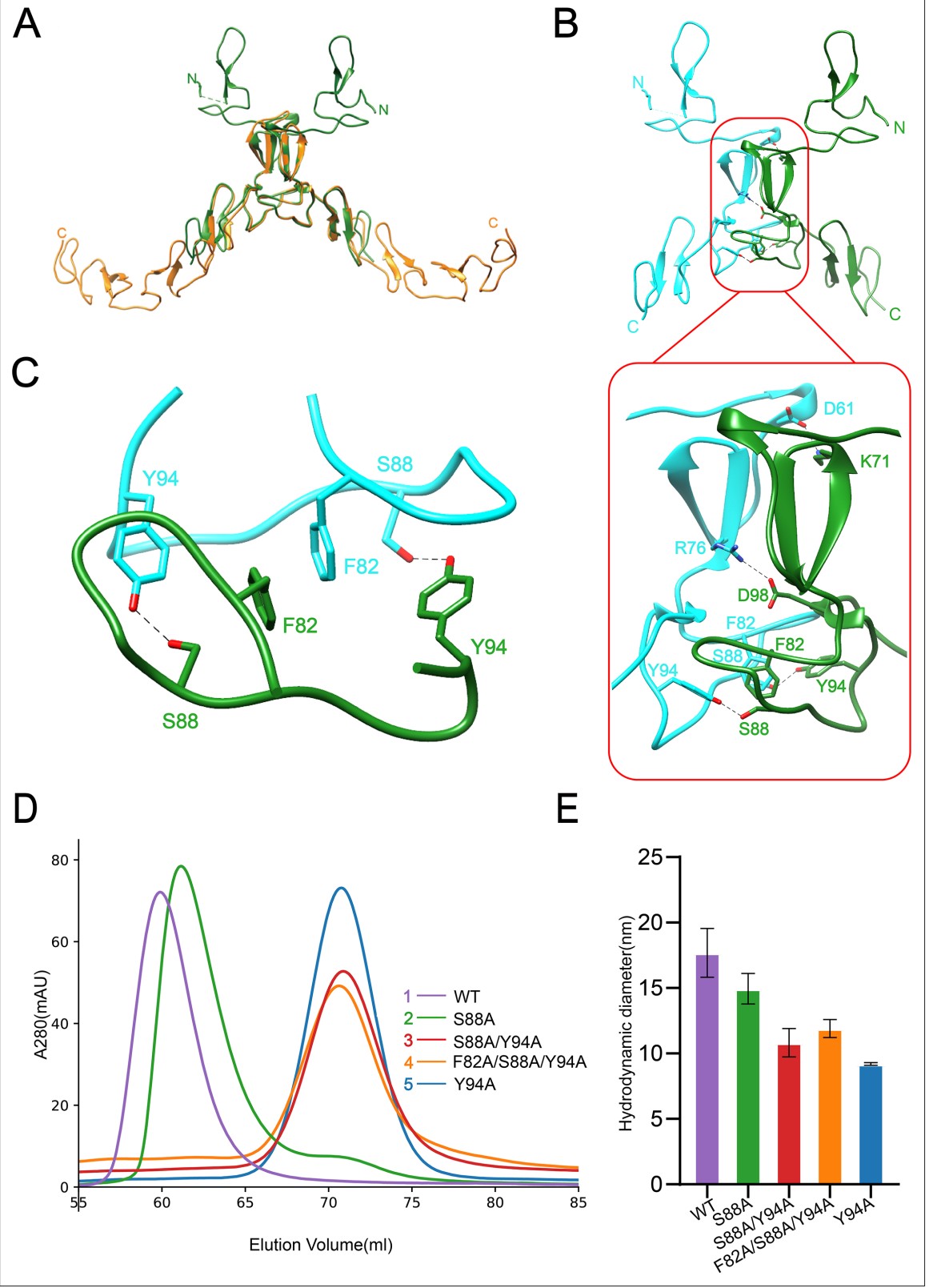

**Figure 2.** Dimerization of SCARF1. (**A**) Superposition of the homodimers of f1 (green) and f2 (gold) in the crystals. (**B**) The dimeric interface of f1 fragment of SCARF1 (red rectangles). Two monomers are colored in cyan and green, respectively. The side chains of the residues that form hydrogen bonds (dashed lines), salt bridges (dashed lines), and π-π interactions are labeled. (**C**) A local view of the dimeric interface of SCARF1. The side chains of the residues that form hydrogen bonds (dashed lines) and π-π interactions are labeled. (**D**) The size exclusion chromatography (SEC) profiles of the wild

*Figure 2 continued on next page*

*Figure 2 continued*

type and the mutants of SCARF1 ectodomain. (**E**) The hydrodynamic diameters of the wild type and the mutants of SCARF1 ectodomain measured by dynamic light scattering (DLS).

The online version of this article includes the following source data and figure supplement(s) for figure 2:

**Figure supplement 1.** Electron density of the crystals and surface electrostatic potential of SCARF1 and SCARF2.

**Figure supplement 2.** Dimerization of SCARF1.

**Figure supplement 2—source data 1.** The original gel displayed in *Figure 2—figure supplement 2A*.

**Figure supplement 2—source data 2.** PDF file containing the original gel for *Figure 2—figure supplement 2A* with labeled bands.

dimerization occurs in a similar fashion for the two fragments in different crystal forms. At the dimeric interfaces of f1 and f2 fragments, hydrogen bonds are formed between S88 and Y94 of the monomers, and F82 and Y94 are also close to each other, thus may form π-π interaction (*Figure 2B and C*, *Figure 2—figure supplement 1A*). Two salt bridges are also observed in f1 crystals between two monomers, one is between D61 and K71, the other is between R76 and D98 (*Figure 2B*). But the salt bridges are not found in f2 crystals, suggesting they are not required for dimerization.

To further characterize the dimerization of SCARF1, a number of mutants of the ectodomain were constructed, and the size exclusion chromatography (SEC) was applied to monitor the elution volumes of the proteins (*Figure 2D*, *Figure 2—figure supplement 2A*). The results showed that mutant S88A, where the hydrogen bonds between S88 and Y94 were removed (*Figure 2C*), had a small elution volume shift compared to the wild type (*Figure 2D*), suggesting that the homodimers were still maintained, but instability may increase for the dimer. By contrast, mutant Y94A, where the hydrogen bonds and the π-π interactions between Y94 and F82 were both removed, showed a significant elution volume shift (*Figure 2D*) and the volume corresponded to the molecular weight of the monomeric SCARF1 ectodomain, suggesting that the π-π interactions between F82 and Y94 were important for dimerization. Other mutants such as S88A/Y94A and F82A/S88A/Y94A also eluted at the volume of the monomer (*Figure 2D*). In parallel, we measured the hydrodynamic diameters of the mutants by dynamic light scattering (DLS), and it showed that the wild type and mutant S88A had larger diameters than mutants Y94A, S88A/Y94A, and F82A/S88A/Y94A (*Figure 2E*, *Figure 2—figure supplement 2B*), which was consistent with the SEC data. In addition, both the dimeric wild type and the monomeric mutant (S88A/Y94A) of the ectodomain exhibited similar SEC elution volumes at pH 6.0 and pH 8.0, suggesting that pH did not have large impact on the dimerization or the conformation of SCARF1 (*Figure 2—figure supplement 2C and D*).

## Interactions of SCARF1 with modified LDLs

To characterize the interaction of SCARF1 with lipoproteins, we monitored the binding of lipoproteins with the SCARF1-transfected HEK293 cells using flow cytometry. The results showed that the SCARF1-transfected cells only bound OxLDL or AcLDL, rather than native LDL (*Figure 3A*, *Figure 3—figure supplement 1*), consistent with the previous reports (*Adachi et al., 1997*; *Wicker-Planquart et al., 2020*), and SCARF1 appeared to have higher binding activity for OxLDL than AcLDL (*Figure 3A*). In parallel, we also tested the interaction of SCARF1 with high-density lipoproteins (HDLs) and oxidized HDL (OxHDL), the results showed that both HDL and OxHDL were not able to bind the SCARF1-transfected cells (*Figure 3A*). Fluorescent confocal images also confirmed that SCARF1 colocalized with OxLDL or AcLDL, rather than LDL, HDL, or OxHDL in the transfected cells (*Figure 3D*). Moreover, the flow cytometry data showed that the binding between SCARF1 and modified LDLs occurred in the presence of Ca$^{2+}$ or EDTA/EGTA (*Figure 3B and C*, *Figure 3—figure supplement 2*), suggesting that the interaction of SCARF1 with OxLDL or AcLDL is Ca$^{2+}$-independent.

To identify the lipoprotein binding region on the ectodomain of SCARF1, we generated a series of truncation mutants, including SCARF1$^{\Delta20-132aa}$, SCARF1$^{\Delta24-221aa}$, SCARF1$^{\Delta222-353aa}$, and SCARF1$^{\Delta353-415aa}$ (*Figure 3E*). The cells transfected with these mutants were applied to examine the interactions with modified LDLs by flow cytometry. The results showed that the cells transfected with SCARF1$^{\Delta20-132aa}$ and SCARF1$^{\Delta222-353aa}$ could bind to OxLDL, similar to the cells expressing the wild type (*Figure 3F–G*). By contrast, the cells expressing SCARF1$^{\Delta24-221aa}$ almost lost binding with OxLDL completely, suggesting that the binding site might locate at the middle region (133–221aa) of the ectodomain of SCARF1 (*Figure 3F*). In addition, the cells expressing SCARF1$^{\Delta353-415aa}$ showed reduced binding with OxLDL

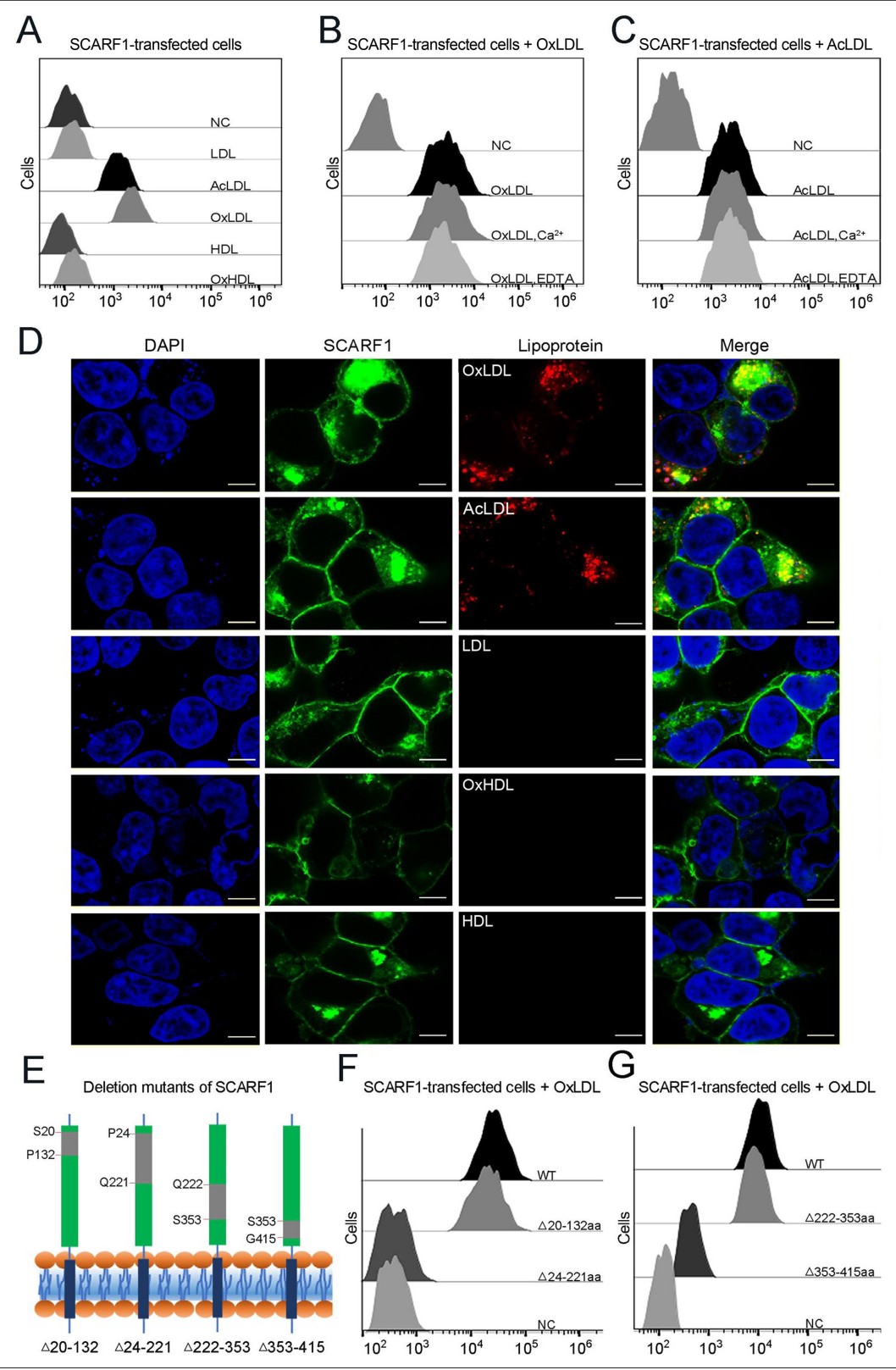

**Figure 3.** SCARF1 recognizes the modified low-density lipoproteins (LDLs). (**A**) Interactions of LDL, acetylated LDL (AcLDL), oxidized LDL (OxLDL), high-density lipoprotein (HDL), and oxidized HDL (OxHDL) with the SCARF1-transfected cells by flow cytometry (NC represents the non-transfected cells and no lipoprotein was added here). (**B**) Interactions of OxLDL with the SCARF1-transfected cells in the presence of Ca²⁺ or EDTA by flow cytometry

*Figure 3 continued on next page*

*Figure 3 continued*

(OxLDL was added for NC). (**C**) Interactions of AcLDL with the SCARF1-transfected cells in the presence of $Ca^{2+}$ or EDTA by flow cytometry (AcLDL was added for NC). (**D**) Confocal fluorescent images of the SCARF1-transfected cells incubated with OxLDL, AcLDL, LDL, OxHDL, or HDL (scale bar, 7.5 µm). (**E**) Schematic diagrams of the deletion mutants of SCARF1. The deleted regions are labeled and shown in gray. (**F**) Interactions of OxLDL with the deletion mutants SCARF1$^{\Delta20-132aa}$ and SCARF1$^{\Delta24-221aa}$ by flow cytometry. (**G**) Interactions of OxLDL with the deletion mutants SCARF1$^{\Delta222-353aa}$ and SCARF1$^{\Delta353-415aa}$ by flow cytometry.

The online version of this article includes the following figure supplement(s) for figure 3:

**Figure supplement 1.** Interaction of oxidized LDL (OxLDL) with the SCARF1-transfected cells by flow cytometry.

**Figure supplement 2.** Interactions of oxidized LDL (OxLDL) with the SCARF1-transfected cells in the presence of $Ca^{2+}$, EDTA, or EGTA by flow cytometry.

**Figure supplement 3.** The expressions of SCARF1 and mutants on the cell surface.

(*Figure 3G*), this may not be surprising as the deletion of the C-terminal regions of the ectodomain might change the conformation of the molecule and generate hinderance for the accessibility of lipo-protein particles.

## SCARF1 recognizes modified LDLs through charge interactions

To identify the binding site for modified LDLs on the region identified above (133–221aa), we calcu-lated the surface electrostatic potential of the region based on the crystal structure of f2 fragment (*Dolinsky et al., 2004*) and found a positively charged area in this region, which is mainly composed of two sites, one is R160 and R161 (Site 1), the other is R188 and R189 (Site 2) (*Figure 4A*). To test whether these positively charged sites are involved in lipoprotein recognition, we generated a number of mutants of SCARF1 including: R160S, R161S, and R160S/R161S for Site 1; R188S, R189S, and R188S/R189S for Site 2; R160S/R161S/R188S, R160S/R188S/R189S, R160S/R161S/R188S/R189S for both sites, and monitored their binding with OxLDL or AcLDL using flow cytometry. The results showed that all the single mutants for the two sites R160S, R161S, R188S, R189S had similar binding activities as the wild type (*Figure 4B and C*), whereas the two double mutants, R160S/R161S for Site 1 or R188S/R189S for Site 2, exhibited reduced binding with modified LDLs (*Figure 4D and E*). And the double mutant for Site 2, R188S/R189S, appeared to have lower binding than the mutant for Site 1, R160S/R161S (*Figure 4D and E*), implying that Site 2, R188S/R189S, may contribute more to the binding. The triple or quadruple mutants, R160S/R161S/R188S, R160S/R188S/R189S, and R160S/R161S/R188S/R189S, where positive charges are largely removed for the two sites, almost lost binding activities with modified LDLs completely (*Figure 4D and E*), suggesting these arginines are crucial for lipoprotein interaction and both sites contribute to the recognition. To further validate the importance of the charged residues, we made mutants where arginines were substituted with lysines on the two sites, including R160K/R161K, R188K/R189K, and R160K/R161K/R188K/R189K. The flow cytometry data showed that all three mutants retained binding activities with modified LDLs (*Figure 4F and G*), confirming the importance of charge interactions. Among them, mutants R160K/R161K and R188K/R189K had similar binding activities as the wild type, and mutant R160K/R161K/R188K/R189K exhib-ited a reduction in lipoprotein binding, suggesting the side chains of lysines may be slightly unfavor-able for the interactions (*Figure 4F and G*). In addition, fluorescent confocal images also confirmed that the mutant R160S/R161S/R188S/R189S did not bind to OxLDL, while the mutants R160K/R161K and R188K/R189K retained binding with OxLDL (*Figure 4H*). Taken together, these data suggest that charge interactions are indispensable for the recognition of modified LDLs by SCARF1.

To confirm the binding position identified above for modified LDLs, we expressed and purified the wild type and mutants of the ectodomain of SCARF1 for ELISA (*Figure 5A*). The results were consis-tent with the flow cytometry data, showing that the mutant of the arginines, R160S/R161S/R188S/R189S, lost binding activity with OxLDL (*Figure 5A*). The ELISA data also showed that the monomeric mutants (S88A/Y94A, F82A/S88A/Y94A) had slightly higher affinities with OxLDL than the dimeric wild type, which might be due to the steric hinderance of the dimers when OxLDLs were coated onto the plates (*Figure 5A*), but flow cytometry suggested that the monomeric mutants had lower binding than the wide type (*Figure 5B*), implying that the dimeric form may be more efficient to recognize lipoproteins on the cell surface. In addition, we also tested the binding of SCARF1 with OxLDL at pH

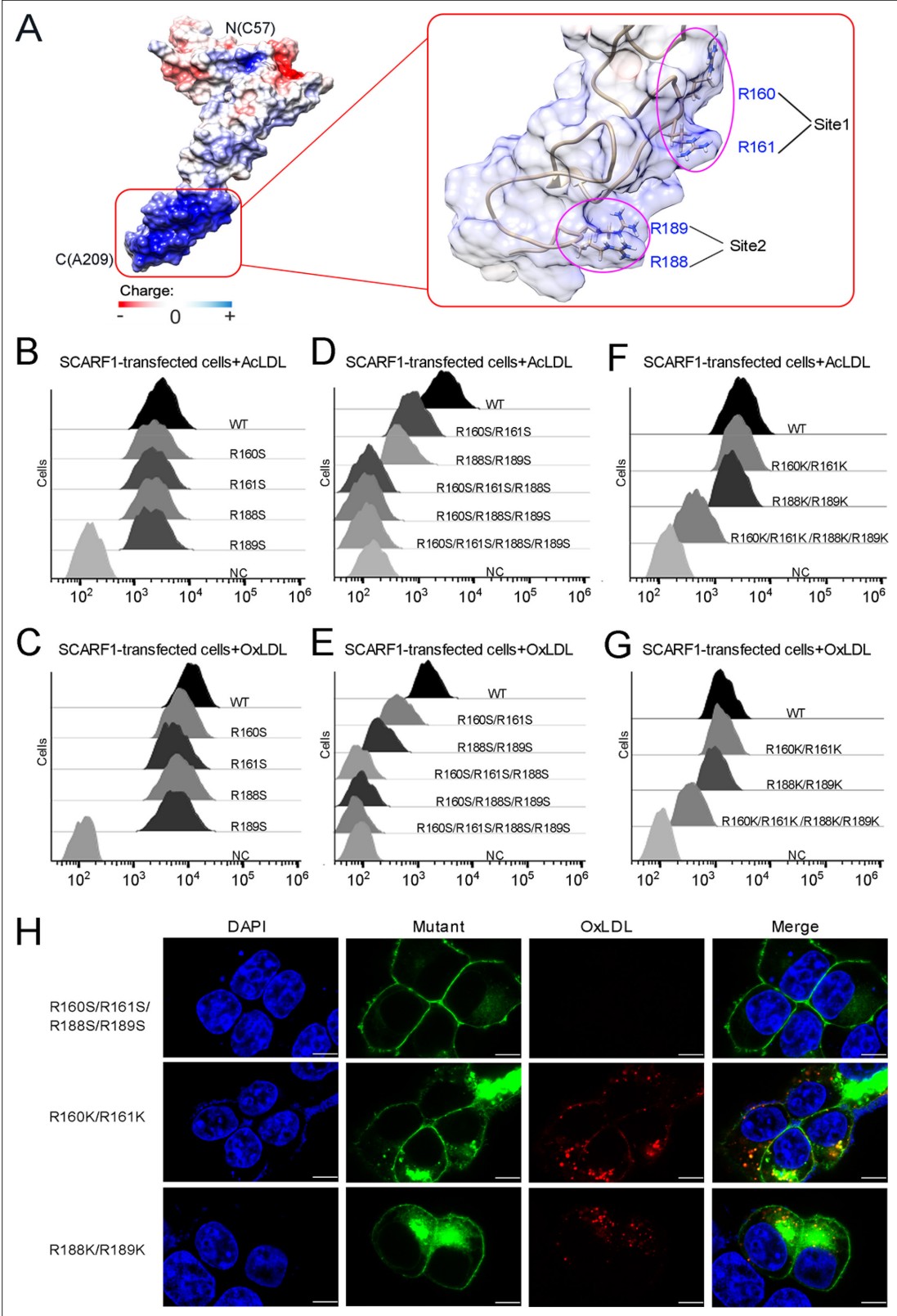

**Figure 4.** The binding sites of modified low-density lipoproteins (LDLs) on SCARF1. (**A**) Surface electrostatic potential of f2 fragment shows a positively charged region on SCARF1 (left, red rectangle), which contains four arginines at Site 1 and Site 2 (right, magenta ovals). (**B**) Interactions of acetylated LDL (AcLDL) with the cells transfected with the single mutants (R to S) of the binding sites by flow cytometry. (**C**) Interactions of oxidized LDL (OxLDL) with the cells transfected with the single mutants (R to S) of the binding sites by flow cytometry. (**D**) Interactions of AcLDL with the cells transfected with

*Figure 4 continued on next page*

*Figure 4 continued*

the double, triple, or quadruple mutants (R to S) of the binding sites by flow cytometry. (**E**) Interactions of OxLDL with the cells transfected with the double, triple, or quadruple mutants (R to S) of the binding sites by flow cytometry. (**F**) Interactions of AcLDL with the cells transfected with the double or quadruple mutants (R to K) of the binding sites by flow cytometry. (**G**) Interactions of OxLDL with the cells transfected with the double or quadruple mutants (R to K) of the binding sites by flow cytometry. (**H**) Confocal fluorescent images of the SCARF1 mutant-transfected cells incubated with OxLDL (scale bar, 7.5 μm).

6.0 by ELISA and flow cytometry, and both data suggested that the binding activity was retained at pH 6.0 (*Figure 5A and B*).

## Interactions of modified LDLs with SCARF1/SCARF2 chimeric molecules

As another member in SR-F class, SCARF2 has similar structural features with SCARF1 according to AlphaFold prediction (*Figure 1—figure supplement 2A and B*). However, the flow cytometry data showed that SCARF2 did not bind to AcLDL or OxLDL (*Figure 6A and B*), which is in agreement with the previous reports (*Ishii et al., 2002*; *Wicker-Planquart et al., 2021*). To further validate the lipoprotein binding sites identified on SCARF1, we generated three pairs of SCARF1/SCARF2 chimeric molecules by switching the counterparts of the two molecules, including: (i) SF1-1 and SF2-1, where 1–421aa of SCARF1 (ectodomain) and 1–441aa of SCARF2 (ectodomain) are switched; (ii) SF1-2 and SF2-2, where 1–221aa of SCARF1 and 1–242aa of SCARF2 are switched; and (iii) SF1-3 and SF2-3, where 133–221aa of SCARF1 and 156–242aa of SCARF2 are switched (*Figure 6C–E*). Both flow cytometry data and fluorescent confocal images showed that SF2-1 gained binding activity with modified LDL, whereas SF1-1 lost the affinity, suggesting that the ectodomain of SCARF1 is sufficient for lipoprotein interaction (*Figure 6C and F–G*). Furthermore, the flow cytometry results of SF1-2, SF2-2, SF1-3, and SF2-3 demonstrated that SCARF2 chimeric molecules could bind modified LDLs when its counterparts are replaced by the fragments from SCARF1 that contain the binding sites (*Figure 6D–F*), thereby confirming the lipoprotein binding sites identified on SCARF1. However, the chimeric molecules SF2-2 and SF2-3 showed weaker affinities with OxLDL than SCARF1 or SF2-1 (*Figure 6F*), suggesting the overall conformation or residues around the substituted binding region of SCARF2 might also affect ligand interaction. In addition, we made a sequence alignment of SCARF1 from different species and found that the positively charged residues at both Site 1 and Site 2 are well

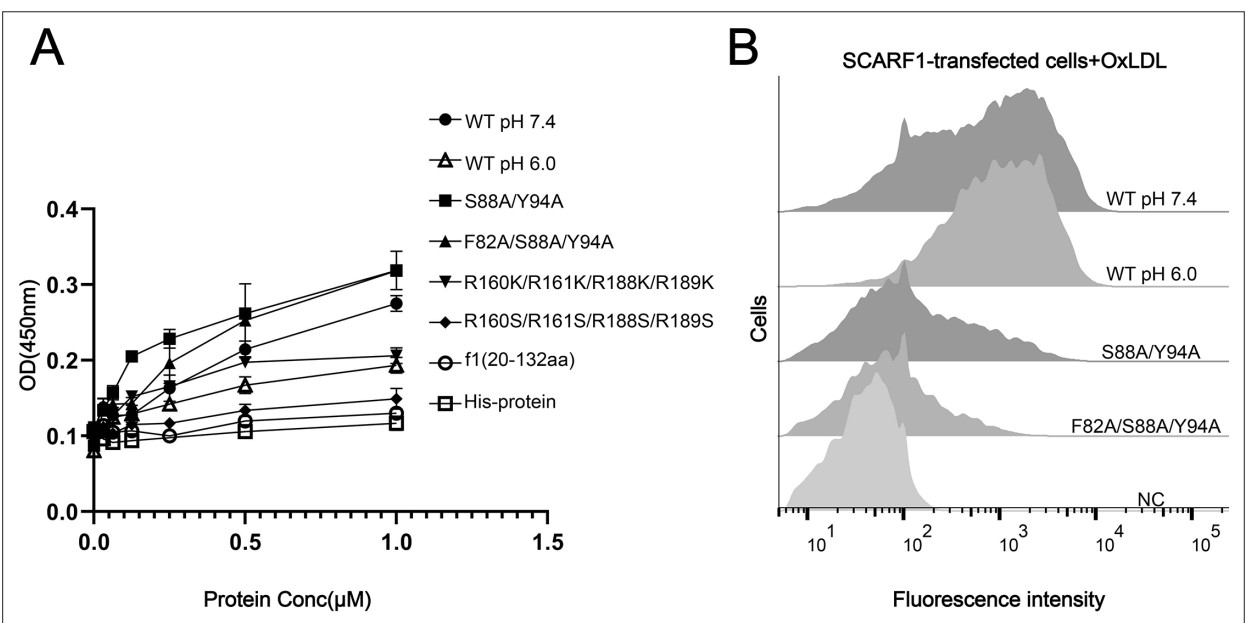

**Figure 5.** The binding of SCARF1 with oxidized LDL (OxLDL). (**A**) ELISA of the interactions of OxLDL with the wild type and the mutants of the ectodomain of SCARF1(f1 fragment is also applied). The assays are performed at pH 7.4 if not labeled. The ectodomain of human HER3 (His-protein) is applied as a control. (**B**) Interactions of OxLDL with the cells transfected with the wild type and the mutants of SCARF1 by flow cytometry. The assays are performed at pH 7.4 if not labeled.

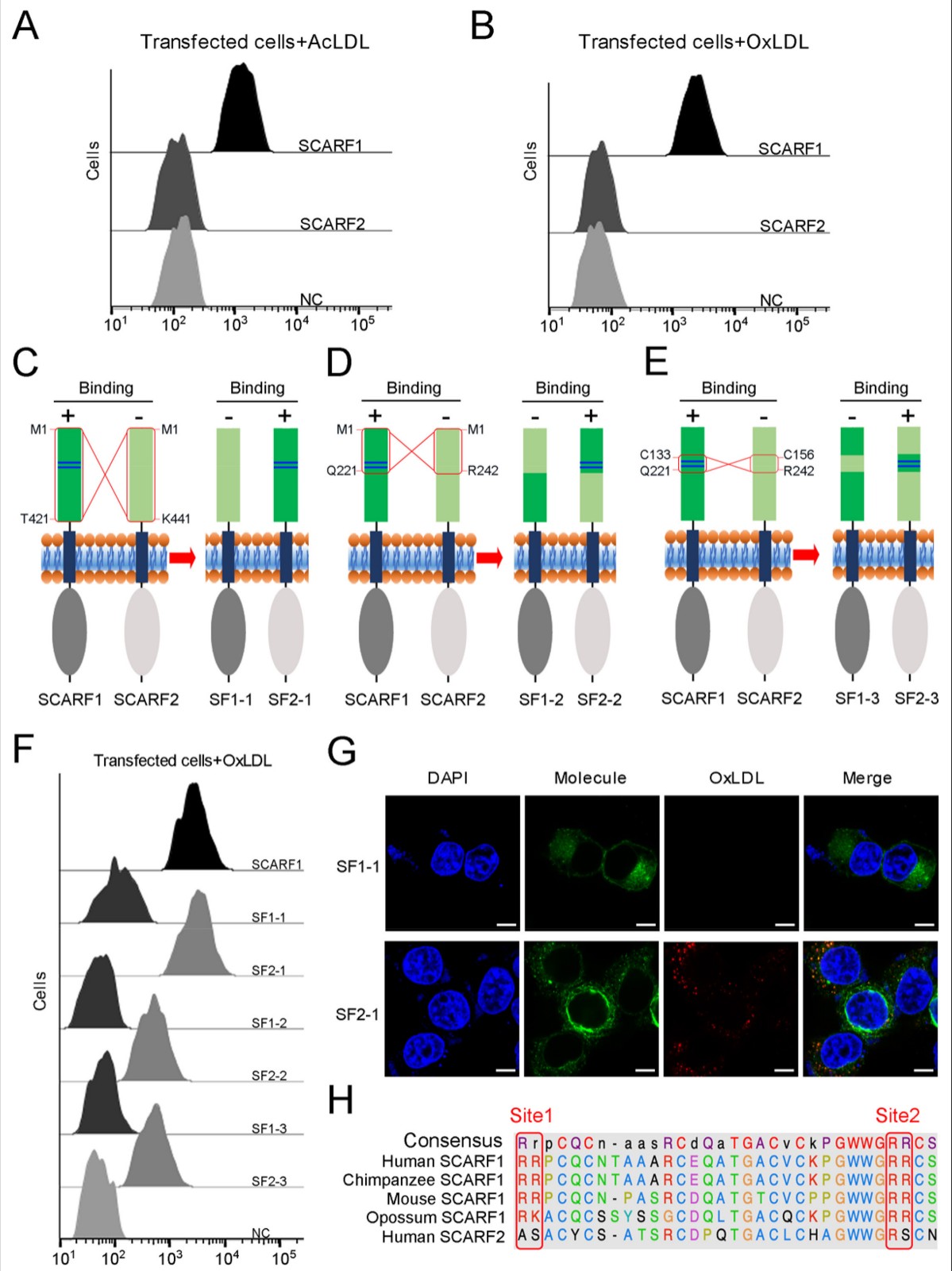

**Figure 6.** The interactions of modified low-density lipoproteins (LDLs) with SCARF1-SCARF2 chimeric molecules. (**A**) Interaction of acetylated LDL (AcLDL) with the SCARF1- or SCARF2-transfected cells by flow cytometry. (**B**) Interaction of oxidized LDL (OxLDL) with the SCARF1- or SCARF2-transfected cells by flow cytometry. (**C**), (**D**), and (**E**) Schematic diagrams of the SCARF1-SCARF2 chimeric molecules generated for binding assays. The switched regions are indicated by red rectangles. The positively charged Site 1 and Site 2 of SCARF1 are shown as blue lines. The binding of molecules

*Figure 6 continued on next page*

*Figure 6 continued*

with OxLDL is indicated as + (positive) or – (negative). (**F**) Interactions of OxLDL with the chimeric molecule-transfected cells by flow cytometry. (**G**) Confocal fluorescent images of the chimera molecule SF1-1- or SF2-1-transfected cells incubated with OxLDL (scale bar, 7.5 µm). (**H**) Sequence alignment of the binding sites of SCARF1 from different species and human SCARF2.

conserved, but not for SCARF2 (*Figure 6H, Figure 2—figure supplement 1B–D*), consistent with the importance of these charged residues in ligand recognition for SCARF1.

## Interaction of SCARF1 with modified LDLs is inhibited by teichoic acids

Previous reports have shown that SCARF1 has multiple ligands (*Adachi et al., 1997*; *Berwin et al., 2004*; *Moriguchi et al., 2018*; *Pfistershammer et al., 2008*; *Hölzl et al., 2011*). Among them, teichoic acids from *Staphylococcus aureus* has been shown to be able to bind SCARF1 in a charge-dependent manner and mediate adhesion to nasal epithelial cells in vitro, and the binding site of teichoic acids locates at the middle region (137–250aa) of the SCARF1 ectodomain (*Baur et al., 2014*). Here, we found that the binding of SCARF1 with OxLDL or AcLDL could be inhibited by teichoic acids efficiently (*Figure 7A and B*). And the two double mutants of SCARF1, where the lipoprotein binding sites are mutated, showed more inhibitory effects from teichoic acids (*Figure 7A and B*), suggesting that the binding sites for modified LDLs and teichoic acids might be shared or overlapped with each other on the ectodomain of SCARF1.

## Discussion

SCARF1 can bind both endogenous and exogenous ligands through its ectodomain (*Murshid et al., 2015*; *Patten, 2018*), which adopts a long-curved conformation with multiple EGF-like domains arranged in tandem according to the crystal structures and the AlphaFold model. EGF-like domains are commonly found in cell surface molecules (*Li et al., 2017*; *Singh et al., 2016*) and usually bind ligands in a $Ca^{2+}$-dependent manner (*Ohlin et al., 1988*; *Dahlbäck et al., 1990*; *Stenflo et al., 2000*). The EGF-like domains of SCARF1 do not contain the typical $Ca^{2+}$-binding sites according to sequence analysis (*Wicker-Planquart et al., 2020*) and no obvious $Ca^{2+}$ density is observed in the

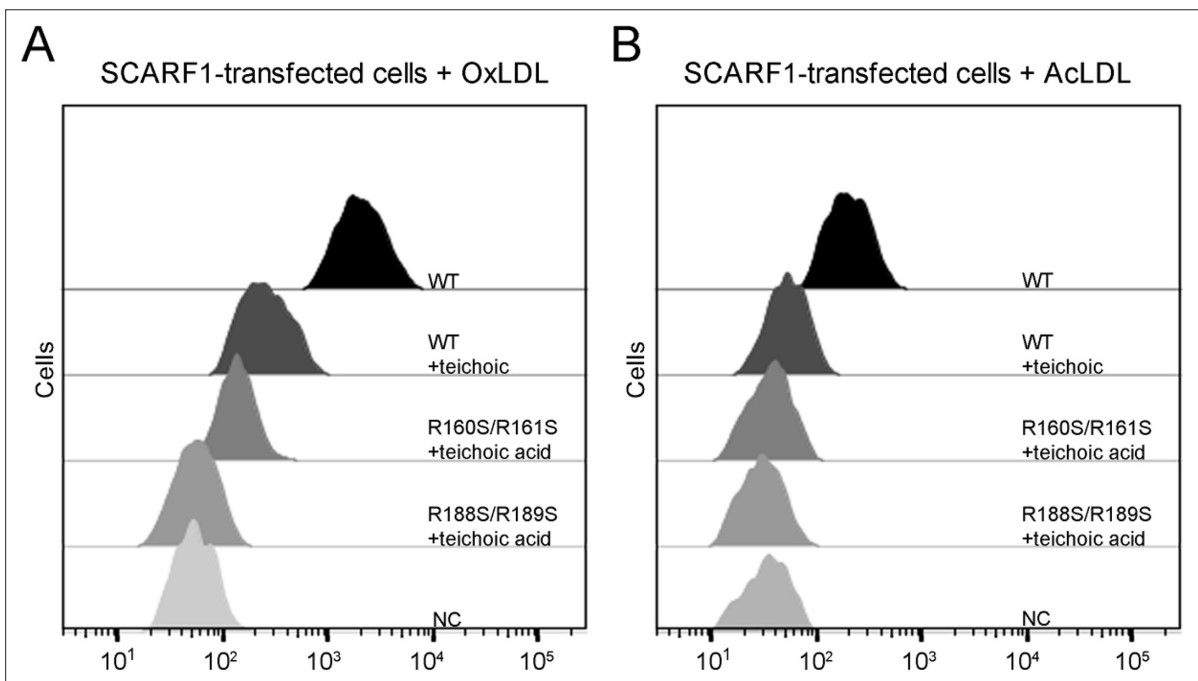

**Figure 7.** Inhibition of the interactions of SCARF1 with modified low-density lipoproteins (LDLs) by teichoic acids. (**A**) Interaction of oxidized LDL (OxLDL) with the cells transfected with the wild type or mutants of SCARF1 in the presence of teichoic acids by flow cytometry. (**B**) Interaction of acetylated LDL (AcLDL) with the cells transfected with the wild type or mutants of SCARF1 in the presence of teichoic acids by flow cytometry.

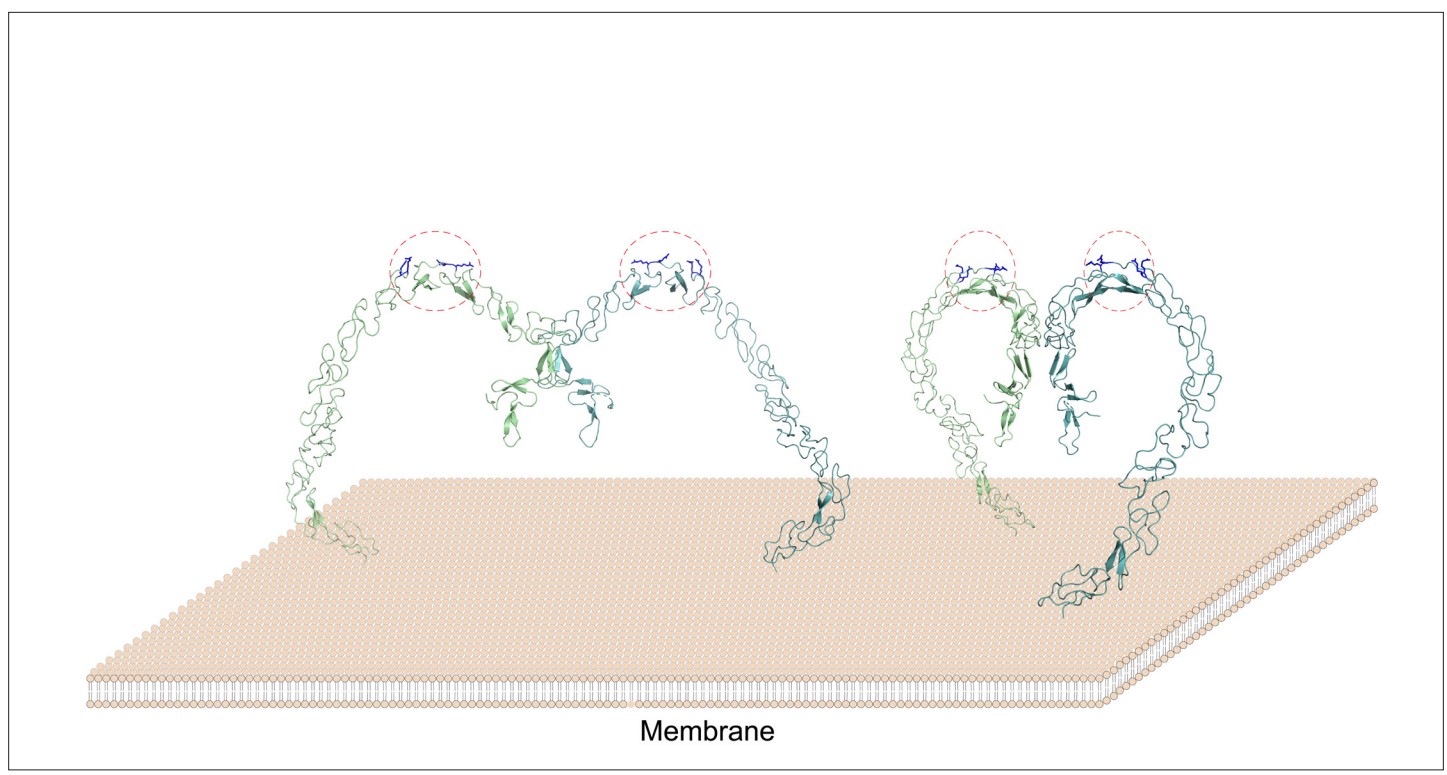

**Figure 8.** A model of the SCARF1 homodimers on the membrane surface. The structure of the SCARF1 homodimer is generated by combining the crystal structures of f1 and f2, and rest of the ectodomain is from AlphaFold prediction. Two dimers are shown with different viewing angles. The monomers are colored in green or cyan. The lipoprotein binding sites are labeled by red dashed ovals and the arginines at the binding sites are colored in blue.

crystal structures, consistent with the results of biochemical assays, suggesting that the interaction of SCARF1 with modified LDLs is $Ca^{2+}$-independent. This is in contrast to the SR-A family, where SCARA1, MARCO, and SCARA5 bind modified LDLs in a $Ca^{2+}$-dependent manner (*Cheng et al., 2021*).

The lipoprotein binding sites of SCARF1 locate at the middle region of the ectodomain and arginines are crucial for recognizing modified LDL by providing positive charges, which may have similarities with WIF-1, a Wnt inhibitory factor containing EGF-like domains that bind glycosaminoglycans through conserved arginines and lysines (*Malinauskas et al., 2011*). In fact, the binding of lipoproteins via charge interactions has been reported before. For LOX-1, which is also an SR, positively charged residues such as arginines play essential roles for its binding with OxLDL (*Shi et al., 2001*; *Ohki et al., 2005*; *Ishigaki et al., 2005*). And similarly, positively charged residues are important to recognize OxLDL by CD36, which is a member in SR-B (*Kar et al., 2008*; *Febbraio et al., 2001*). Therefore, the interaction of SCARF1 with modified LDLs might be similar to LOX-1 or the SR-B members, rather than the SR-A members. Moreover, both structural and biochemical data suggest that SCARF1 forms homodimers (*Figure 8*), which may facilitate the accessibility or binding of modified lipoproteins on the cell surface, and is also analogous to some of the lipoprotein receptors such as LOX-1 (*Park et al., 2005*; *Gaidukov et al., 2011*).

Among the ligands of SCARF1, the teichoic acids from *S. aureus* have been shown to interact with SCARF1 in a charge-dependent manner (*Baur et al., 2014*), which is similar to modified LDLs that also need charged residues to bind SCARF1. Therefore, teichoic acids and modified LDLs may share the binding sites on the ectodomain of SCARF1, implying that this region might be a general binding position for some of the scavenging targets, but whether all the ligands utilize this region for binding needs further investigation. In addition, the sequence alignment also shows that these charged residues are well conserved in different species for SCARF1, but SCARF2 lacks such sites on the ectodomain, suggesting SCARF1 and SCARF2 are functionally divergent, although they have some sequence similarities.

Over the past decades, SCARF1 has been linked to a number of diseases, including cardiovascular diseases, systemic lupus erythematosus, fungal keratitis, and cancer (*Patten et al., 2020*; *Jorge et al., 2022*; *Patten, 2018*; *Zhang et al., 2021*). But the mechanisms and roles of SCARF1 in these diseases are largely elusive. The structural and mechanistic characterization of SCARF1 and its interactions of multiple ligands would shed light on the physiological and pathological roles of this receptor on the corresponding pathways and may also provide insights on the therapeutic strategies against the related diseases.

# Materials and methods

## Protein expression and purification

For protein expression in insect cells, the cDNA of human SCARF1 (NCBI nm_003693) encoding f1 (20–132aa) and f2 (20-221aa) fragments were sub-cloned into a pFastBac vector fused with an N-terminal melittin signal peptide and a C-terminal 6xHis tag using NovoRec recombinant enzyme. Sf9 cells were used for generating recombinant baculoviruses and high five cells were used for protein production. The infected cells were cultured in ESF921 medium (Expression Systems) for 3 days in a 27°C humidified incubator. The supernatants were buffer-exchanged with 25 mM Tris, 150 mM NaCl at pH 8.0 by dialysis, then applied to Ni-NTA chromatography (Ni-NTA Superflow, QIAGEN). The eluted proteins were concentrated to 1 ml using Amicon Ultra-15 3k-cutoff filter (Millipore) and loaded onto a HiLoad Superdex 75 prep grade column or a HiLoad Superdex 200 prep grade column (GE Healthcare) with Tris-NaCl buffer (10 mM Tris, 150 mM NaCl at pH 7.5) for further purification. The purified proteins were loaded onto SDS-PAGE (12%) and stained with coomassie brilliant blue R250 for detection.

For protein expression in mammalian cells, the cDNAs of the human SCARF1 ectodomain and mutants (S88A, Y94A, S88A/Y94A, F82A/S88A/Y94A) were sub-cloned into a pTT5 vector fused with a C-terminal 6xHis tag. The transfected HEK293F cells were cultured in suspension for 5 days in a $CO_2$ humidified incubator at 37°C, then the supernatants were collected and applied to Ni Smart Beads (Smart-Lifesciences). The eluted proteins were concentrated to 1 ml using Amicon Ultra-15 10k-cutoff filter (Millipore) and loaded onto a HiLoad Superdex 200 prep grade column (GE Healthcare) with Tris-NaCl buffer (50 mM Tris, 150 mM NaCl at pH 8.0) for further purification. The purified proteins were loaded onto SDS-PAGE (10%) and stained with coomassie brilliant blue R250 for detection.

## Cell culture and transfection

HEK293T cells were cultured in Dulbecco's modified Eagle's medium (HyClone) supplemented with 10% fetal bovine serum and cells were incubated in a humidified incubator at 37°C with 5% $CO_2$. The cells were tested negative for mycoplasma contamination. The cells were seeded in a 12-well plate in advance, and on the day of transfection, the cells that were 80–90% confluent in the 12-well plate were transfected using PEI (Polyplus). After 4–6 hr of transfection, the culture medium was replaced by the fresh medium. Then, about 24 hr after transfection, the cells were ready for assays.

HEK293F cells were cultured in suspension with Union-293 medium (Union Bio) in a humidified incubator at 37°C with 8% $CO_2$. Transfections were done at the cell density of 1.5–2.0×$10^6$/ml using PEI (Polyplus) and after 24 hr, the cells were ready for assays.

## Crystallization and structural determination

The purified protein was concentrated to 30 mg/ml (f1, 20–132aa) and 10 mg/ml (f2, 20–221aa) (measured by UV absorption at 280 nm) using Amicon Ultra-15 3k-cutoff filter (Millipore). Crystal screening was done at 18°C by sitting-drop vapor diffusion method using 96-well plates (Swissci) with commercial screening kits (Hampton Research). A Mosquito nanoliter robot (TTP Labtech) was used to set up 200 nl protein sample mixed with 200 nl reservoir solution. The crystals of f1 were grown in a solution containing 20% (wt/vol) polyethylene glycol 3350, 0.2 M ammonium sulfate, 0.1 M Bis-Tris (pH 5.5) and the crystals of f2 were grown in 0.1 M citric acid/sodium citrate (pH 3.6), 0.2 M sodium citrate tribasic, 9% PEG6000. Crystals of both fragments were obtained after 5–7 days. 10% ethylene glycol was added during crystal harvest and data collection as cryo-protectant. The f1 crystal heavy atom derivatives were obtained by soaking with 10 mM $K_2PtCl_4$ for 2 min before data collection. Diffraction data were collected using a PILATUS 6M detector at BL18U1 beamline of National Facility

for Protein Science Shanghai (NFPS) at Shanghai Synchrotron Radiation Facility (SSRF). X-ray diffraction data were integrated and scaled with HKL-3000 package (*Minor et al., 2006*).

The initial phasing of the f1 crystals was done by SAD with Pt derivatives, then the structure was used as a search model of molecular replacement for a native dataset of f1 with higher resolution for further refinement. The f2 crystal structure was solved by molecular replacement using the structure of f1 fragment combined with the EGF-like fragment predicted by AlphaFold as search models using Phenix (*Liebschner et al., 2019*; *McCoy et al., 2007*). Model building and refinement were done using Coot (*Emsley and Cowtan, 2004*) and Phenix (*Afonine et al., 2012*; *Williams et al., 2018*). The structural figures were prepared using UCSF Chimera (*Pettersen et al., 2004*). The coordinates and the structure factors have been deposited in the Protein Data Bank with entry 8HN0 and 8HNA for f1 and f2 fragments, respectively.

## Mutagenesis experiments

SCARF1 mutations, including base-substitution mutations (R160S, R161S, R188S, R189S, R160S/ R161S, R188S/R189S, R160K/R161K, R188K/R189K, R160S/R161S/R188S, R160S/R188S/R189S, R160S/R161S/R188S/R189S, R160K/R161K/R188K/R189K,S88A,Y94A, S88A/Y94A, F82A/S88A/ Y94A), deletion mutations (SCARF1$^{\Delta 20-132aa}$, SCARF1$^{\Delta 24-221aa}$, SCARF1$^{\Delta 222-353aa}$, SCARF1$^{\Delta 353-415aa}$) were introduced into the corresponding plasmids by PCR using KOD DNA polymerase (Sparkjade Science Co., Ltd.). The template plasmids were digested by DpnI (Thermo Fisher Scientific), and the digested PCR products were ligated by Ligation High (TOYOBO). The chimeric molecules of SCARF1 and SCARF2 were constructed by replacing fragments (1–441aa, 1–242aa, or 156–242aa) of SCARF2 with its counterparts of SCARF1 (1–421aa, 1–221aa, or 133–221aa), respectively.

## Dynamic light scattering

The purified protein samples were concentrated to about 100 µg/ml in Tris buffer (50 mM Tris, 150 mM NaCl at pH 8.0) and DLS signals were measured and processed on a Zetasizer Pro analyzer (Malvern Panalytical). Data for each sample were collected in triplicate at 25°C.

## Preparation of modified lipoproteins

Lipoproteins (purity, 97–98%), including Dil-LDL (20614ES76), Dil-AcLDL (20606ES76), Dil-OxLDL (20609ES76), LDL (20613ES05), AcLDL (20604ES05) were purchased from Yeasen. Dil-HDL (H8910) was purchased from Solarbio. The lipoproteins mentioned above (LDL, AcLDL, OxLDL, HDL) were isolated from human plasma.

For preparation of OxHDL, 140 µl of 1 mg/ml HDL were buffer-exchanged to PBS solution, and then the same volume of 100 µM CuSO$_4$ was divided into multiple small portions and added to HDL solutions gradually. After 16 hr at 37°C, the reaction solutions were dialyzed against the PBS buffer containing 0.3 mM EDTA to stop the reaction.

## Flow cytometry

For the binding assays of SCARF1 and SCARF2 with lipoproteins, HEK293T cells were transiently transfected with the full-length SCARF1 or SCARF2 fused with a C-terminal GFP tag. After 24 hr, 5 µg Dil-tagged (wavelength: 565 nm) lipoprotein (Dil-LDL, Dil-AcLDL, Dil-OxLDL, Dil-HDL, Dil-OxHDL) was added to the culture medium. After 2–4 hr at 37°C, cells were washed three times with the washing buffer (25 mM HEPES, 150 mM NaCl, 0.1% Tween 20, pH 7.4) and then washed twice with the cleaning buffer (25 mM HEPES, 150 mM NaCl, pH 7.4) for flow cytometry.

For the Ca$^{2+}$ assays, HEK293T cells were transiently transfected with the full-length SCARF1 fused with a C-terminal GFP tag. After 24 hr, 5 µg Dil-labeled lipoprotein (Dil-AcLDL, Dil-OxLDL) was added to the culture medium containing 2 mM Ca$^{2+}$ or 2 mM EDTA/EGTA. After 2–4 hr, the cells were washed twice with the corresponding washing buffer (25 mM HEPES, 150 mM NaCl, 0.1% Tween 20, 2 mM Ca$^{2+}$, or 2 mM EDTA/EGTA, pH 7.4) and then washed twice with the cleaning buffer (25 mM HEPES, 150 mM NaCl, 2 mM Ca$^{2+}$, or 2 mM EDTA/EGTA, pH 7.4) for flow cytometry.

For the binding assays of the SCARF1 mutants with lipoproteins, HEK293T cells were transiently transfected with the wild type or the mutants of SCARF1 fused with a C-terminal GFP tag. After 24 hr, 5 µg Dil-tagged (wavelength: 565 nm) lipoprotein (Dil-LDL, Dil-AcLDL, Dil-OxLDL) was added to the culture medium. After 2–4 hr at 37°C, cells were washed three times with the washing buffer (25 mM

HEPES, 150 mM NaCl, 0.1% Tween 20, pH 7.4) and then washed twice with the cleaning buffer (25 mM HEPES, 150 mM NaCl, pH 7.4) for flow cytometry. For the assays performed at pH 6.0, PBS buffer (pH 6.0) was used following the similar procedure.

For the teichoic acids assays, HEK293T cells were transiently transfected with the full-length SCARF1 fused with a C-terminal GFP tag. After 24 hr, teichoic acids (100 µg/ml) were added to the culture medium. After 1–2 hr at 4°C, 5 µg Dil-labeled lipoprotein (Dil-AcLDL, Dil-OxLDL) was added to the culture medium. After 2–4 hr at 37°C, cells were washed three times with the washing buffer (25 mM HEPES, 150 mM NaCl, 0.1% Tween 20, pH 7.4) and then washed twice with the cleaning buffer (25 mM HEPES, 150 mM NaCl, pH 7.4) for flow cytometry.

For the assays to monitor the expression of SCARF1 and mutants, HEK293F cells were transiently transfected with the wild type or the mutants fused with a C-terminal GFP tag. After 24 hr, Human SREC-I/SCARF1 Alexa Fluor 647-conjugated Antibody (FAB2409R, R&B SYSTEMS) was added to the culture medium. After incubation of 30 min at 4°C, cells were washed three times with the washing buffer (135 mM NaCl, 2.7 mM KCl, 1.5 mM $KH_2PO_4$, and 8 mM $K_2HPO_4$) for flow cytometry (*Figure 3— figure supplement 3*).

Flow cytometry data were acquired using an LSR Fortessa flow cytometer (BD Biosciences). Data analysis was performed using FlowJo software (Tree Star, Inc).

## ELISA experiments

Lipoprotein (OxLDL) was coated onto 96-well plates with 1 µg protein per well at 4°C overnight. The plates were then blocked with the blocking buffer (25 mM HEPES, 150 mM NaCl, 0.1% Triton X-100, and 5% [wt/vol] BSA, pH 7.4) at room temperature for 3 hr. The purified His-tagged proteins was serially diluted and added to each well in the binding buffer (25 mM HEPES, 150 mM NaCl, 0.1% Triton X-100, and 2 mg/ml BSA, pH 7.4). After incubation at room temperature for 2 hr, the plates were washed five times with the washing buffer (25 mM HEPES, 150 mM NaCl, and 0.1% Triton X-100, pH 7.4) and then incubated with the mouse anti-His antibody (66005-1-Ig, Proteintech) for 1 hr. After washing three times with the washing buffer, the plates were incubated with the HRP-labeled goat anti-mouse IgG (A0216, Beyotime) for 1 hr. After washing three times with the washing buffer, 100 µl TMB Chromogen Solution (P0209-500ml, Beyotime) was added to each well and incubated for 30 min at room temperature. Then, 50 µl $H_2SO_4$ (2.0 M) was added to each well to stop the reactions. For the binding of lipoproteins at pH 6.0, PBS buffer (pH 6.0) was used following the similar procedure. The plates were then read at 450 nm on a Synergy2 machine (Bio Tek Instruments). The ELISA data shown in the figures are representative of three repeated experiments and presented as mean ± SD.

## Confocal microscopy

HEK293T cells were grown on coverslips and transfected with the full-length SCARF1, SCARF2, or the mutants of SCARF1 fused with a C-terminal GFP tag using six-well plates. After 24 hr of transfection, 5 µg Dil-labeled lipoprotein (Dil-LDL, Dil-AcLDL, Dil-OxLDL, Dil-OxHDL) added to the plates. After 2–4 hr of incubation, the cells were fixed by 4% paraformaldehyde in TBS (50 mM Tris and 150 mM NaCl, pH 7.4). After washing twice with the buffer (25 mM HEPES, 150 mM NaCl, 0.05% Tween-20, pH 7.4), the cells were blocked in the blocking buffer (25 mM HEPES, 150 mM NaCl, 5% [wt/vol] BSA, 0.05% Tween-20, pH 7.4) at room temperature for 1 hr. After washing three times with the washing buffer (25 mM HEPES, 150 mM NaCl, 0.05% Tween-20, pH 7.4), the cells were incubated with 5 µM DAPI for 15 min. Then, the plates were washed again for confocal microscopy with a Leica SP8 microscope.

## Acknowledgements

We thank the Integrated Laser Microscopy System and the Large-scale Protein Production System at the National Facility for Protein Science in Shanghai (NFPS), Shanghai Advanced Research Institute, Chinese Academy of Sciences, China, for technical support. We also thank the beamline BL18U1 of National Facility for Protein Science Shanghai (NFPS) at Shanghai Synchrotron Radiation Facility for their assistance in X-ray diffraction data collection. This work is supported by National Natural Science Foundation of China (No. 91957102) to YH and we also thank the support from innovative research team of high-level local universities in Shanghai (SHSMU-ZLCX20212601).

## Additional information

### Funding

| Funder | Grant reference number | Author |
|---|---|---|
| National Natural Science Foundation of China | 91957102 | Yongning He |
| Innovative Research Team of High-level Local Universities in Shanghai | SHSMU-ZLCX20212601 | Yongning He |

The funders had no role in study design, data collection and interpretation, or the decision to submit the work for publication.

### Author contributions

Yuanyuan Wang, Fan Xu, Validation, Investigation, Methodology, Writing – original draft, Writing – review and editing; Guangyi Li, Investigation, Methodology; Chen Cheng, Bowen Yu, Ze Zhang, Dandan Kong, Fabao Chen, Yali Liu, Longxing Cao, Investigation; Zhen Fang, Data curation, Investigation; Yang Yu, Yijun Gu, Methodology; Yongning He, Conceptualization, Resources, Supervision, Funding acquisition, Methodology, Project administration, Writing – review and editing

### Author ORCIDs

Fan Xu ⓘ http://orcid.org/0009-0006-3535-1689
Yongning He ⓘ https://orcid.org/0000-0002-1640-5104

Reviewer #1 (Public review): https://doi.org/10.7554/eLife.93428.3.sa1
Author response https://doi.org/10.7554/eLife.93428.3.sa2

## Additional files

### Supplementary files

• Supplementary file 1. X-ray data collection and processing.
• Supplementary file 2. Crystallographic statistics of the structures.
• MDAR checklist

### Data availability

The crystal structures of the human SCARF1 fragments, f1 and f2, are deposited in PDB (https://www.rcsb.org) with entry 8HN0 and 8HNA, respectively.

The following datasets were generated:

| Author(s) | Year | Dataset title | Dataset URL | Database and Identifier |
|---|---|---|---|---|
| Wang Y, Xu F, Li G, Cheng C, Yu B, Zhang Z, Kong D, Chen F, Liu Y, Fang Z, Cao L, Yu Y, Gu Y, He Y | 2024 | Structure of scavenger receptor SCARF1 and its interaction with lipoproteins | https://www.rcsb.org/structure/8HN0 | RCSB Protein Data Bank, 8HN0 |
| Wang Y, Xu F, Li G, Cheng C, Yu B, Zhang Z, Kong D, Chen F, Liu Y, Fang Z, Cao L, Yu Y, Gu Y, He Y | 2024 | Structure of scavenger receptor SCARF1 and its interaction with lipoproteins | https://www.rcsb.org/structure/8HNA | RCSB Protein Data Bank, 8HNA |

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
