## [Editor Report · eLife Assessment]

SCARF1 is a scavenger membrane-bound receptor that binds modified versions of lipoproteins and has a significant role in maintaining lipid homeostasis. This **useful** study reports the crystal structure of SCARF1 and identifies putative binding sites for modified lipoproteins. Supported by a **convincing** set of experimental approaches, this study advances our knowledge of how scavenger receptors clear modified lipoproteins to maintain lipid homeostasis.

---

## [Referee Report · Reviewer #1 (Public review)]

Summary:

This manuscript provides a solid advance to the scavenger receptor field by reporting the crystal structures of the domains of SCARF1 that bind modified LDL such as oxidized LDL and acylated LDL. The crystal packing reveals a new interface for homodimerization of SCARF1. The authors characterize SCARF1 binding to modified LDL using flow cytometry, ELISA, and fluorescent microscopy. They identify a positively-charged surface on the structure that they predict will bind the LDLs, and they support this hypothesis with several mutant constructs in binding experiments.

Strengths:

The authors have crystallized domains of an understudied scavenger receptor and used the structure to identify a putative binding site for modified LDL particles. An especially strong set of experiments are binding studies with chimeras of SCARF1 and SCARF2, where they show gain-of-function results (binding of modified LDLs) by SCARF2, a related protein that does not normally bind modified LDLs. The paper is a straightforward set of experiments that identify the likely binding site of modified LDL on SCARF1

Weaknesses:

In the current revision, the authors addressed my technical concerns.

Two remaining considerations that may limit the broader impact of this paper are (1) that it does not explain the structural basis for specificity of the binding of SCARF1 to various lipoproteins (i.e. why SCARF1 binds oxLDL and AcLDL but not LDL or HDL) and (2) a lack of a biological assay to interpret the functional consequences of the SCARF1 mutants. These may be addressed in future work.

---

## [Author Response]

The following is the authors’ response to the original reviews.

**Public Reviews:**

**Reviewer #1 (Public Review):**
Summary:This study provides an incremental advance to the scavenger receptor field by reporting the crystal structures of the domains of SCARF1 that bind modified LDL such as oxidized LDL and acylated LDL. The crystal packing reveals a new interface for the homodimerization of SCARF1. The authors characterize SCARF1 binding to modified LDL using flow cytometry, ELISA, and fluorescent microscopy. They identify a positively charged surface on the structure that they predict will bind the LDLs, and they support this hypothesis with a number of mutant constructs in binding experiments.Strengths:The authors have crystallized domains of an understudied scavenger receptor and used the structure to identify a putative binding site for modified LDL particles. An especially interesting set of experiments is the SCARF1 and SCARF2 chimeras, where they confer binding of modified LDLs to SCARF2, a related protein that does not bind modified LDLs, and use show that the key residues in SCARF1 are not conserved in SCARF2.Weaknesses:While the data largely support the conclusions, the figures describing the structure are cursory and do not provide enough detail to interpret the model or quality of the experimental X-ray structure data. Additionally, many of the flow cytometry experiments lack negative controls for non-specific LDL staining and controls for cell surface expression of the SCARF constructs. In several cases, the authors interpret single data points as increased or decreased affinity, but these statements need dose-response analysis to support them. These deficiencies should be readily addressable by the authors in the revision.The paper is a straightforward set of experiments that identify the likely binding site of modified LDL on SCARF1 but adds little in the way of explaining or predicting other binding interactions. That a positively charged surface on the protein could mediate binding to LDL particles is not particularly surprising. This paper would be of greater importance if the authors could explain the specificity of the binding of SCARF1 to the various lipoparticles that it does or does not bind. Incorporating these mutants into an assay for the biological role of SCARF1 would be powerful.
**Reviewer #2 (Public Review):**
Summary:The manuscript by Wang and colleagues provided mechanistic insights into SCARF1 and its interactions with the lipoprotein ligands. The authors reported two crystal structures of the N-terminal fragments of SCARF1 ectodomain (ECD). On the basis of the structural analysis, the authors further investigated the interactions between SCARF1 and modified LDLs using cell-based assays and biochemical experiments. Together with the two structures and supporting data, this work provided new insights into the diverse mechanisms of scavenger receptors and especially the crucial role of SCARF1 in lipid metabolism.Strengths:The authors started by determining the crystal structures of two fragments of SCARF1 ECD. The superposition of the two high-resolution structures, together with the predicted model by AlphaFold, revealed that the ECD of SCARF1 adopts a long-curved conformation with multiple EGF-like domains arranged in tandem. Non-crystallographic and crystallographic two-fold symmetries were observed in crystals of f1 and f2 respectively, indicating the formation of SCARF1 homodimers. Structural analysis identified critical residues involved in dimerization, which were validated through mutational experiments. In addition, the authors conducted flow cytometry and confocal experiments to characterize cellular interactions of SCARF1 with lipoproteins. The results revealed the vital role of the 133-221aa region in the binding between SCARF1 and modified LDLs. Moreover, four arginine residues were identified as crucial for modified LDL recognition, highlighting the contribution of charge interactions in SCARF1-lipoprotein binding. The lipoprotein binding region is further validated by designing SCARF1/SCARF2 chimeric molecules. Interestingly, the interaction between SCARF1 and modified LDLs could be inhibited by teichoic acid, indicating potential overlap in or sharing of binding sites on SCARF1 ECD.The author employed a nice collection of techniques, namely crystallographic, SEC, DLS, flow cytometry, ELISA, and confocal imaging. The experiments are technically sound and the results are clearly written, with a few concerns as outlined below. Overall, this research represents an advancement in the mechanistic investigation of SCARF1 and its interaction with ligands. The role of scavenger receptors is critical in lipid homeostasis, making this work of interest to the eLife readership.
**Reviewer #3 (Public Review):**
Summary:The manuscript by Wang et. al. described the crystal structures of the N-terminal fragments of Scavenger receptor class F member 1 (SCARF1) ectodomains. SCARF1 recognizes modified LDLs, including acetylated LDL and oxidized LDL, and it plays an important role in both innate and adaptive immune responses. They characterized the dimerization of SCARF1 and the interaction of SCARF1 with modified lipoproteins by mutational and biochemical studies. The authors identified the critical residues for dimerization and demonstrated that SCARF1 may function as homodimers. They further characterized the interaction between SCARF1 and LDLs and identified the lipoprotein ligand recognition sites, the highly positively charged areas. Their data suggested that the teichoic acid inhibitors may interact with SCARF1 in the same areas as LDLs.Strengths:The crystal structures of SCARF1 were high quality. The authors performed extensive site-specific mutagenesis studies using soluble proteins for ELISA assays and surface-expressed proteins for flow cytometry.Weaknesses:(1) The schematic drawing of human SCARF1 and SCARF2 in Fig 1A did not show the differences between them. It would be useful to have a sequence alignment showing the polymorphic regions.

The schematic drawing in Fig.1A is to give a brief idea about the two molecules, the sequence alignment may take too much space in the figure. A careful alignment between SCARF1 and SCARF2 can be found in Ref. 24 (Ishii, et al., J Biol Chem, 2002. 277, 39696-702) an also mentioned in p.4.

(2) The description of structure determination was confusing. The f1 crystal structure was determined by SAD with Pt derivatives. Why did they need molecular replacement with a native data set? The f2 crystal structure was solved by molecular replacement using the structure of the f1 fragment. Why did they need to use EGF-like fragments predicted by AlphaFold as search models?

The crystal structure of f1 was first determined by SAD using Pt derivatives, but soaking of Pt reduced the resolution of the crystals, therefore we use this structure as a search model for a native data set that had higher resolution for further refinement. For the structural determination of f2, the molecular replacement using f1 structure was not able to show the initial density of the extra region in f2 (residues 133-209), which was missing in f1. Therefore, the EGF-like domains of SCARF1 modeled by AlphaFold were applied as search models for this region (p.18).

(3) It's interesting to observe that SCARA1 binds modified LDLs in a Ca2+-independent manner. The authors performed the binding assays between SCARF1 and modified LDLs in the presence of Ca2+ or EDTA on Page 9. However, EDTA is not an efficient Ca2+ chelator. The authors should have performed the binding assays in the presence of EGTA instead.

The binding assays in the presence of EGTA are included in the revised manuscript (Fig. S7) (p.9), which also suggest that SCARA1 binds OxLDL in a Ca2+-independent manner.

(4) The authors claimed that SCARF1Δ353-415, the deletion of a C-terminal region of the ectodomain, might change the conformation of the molecule and generate hinderance for the C-terminal regions. Why didn't SCARF1Δ222-353 have a similar effect? Could the deletion change the interaction between SCARF1 and the membrane? Is SCARF1Δ353-415 region hydrophobic?

The truncation mutants were constructed to roughly locate the binding region of lipoproteins on SCARF1, and the overall results showed that the sites might locate at the region of 133-221. Mutant Δ222-353 may also affect the conformation, but it still had binding with OxLDL like wild type, suggesting the binding sites were retained in this mutant. Mutant Δ353-415 showed a reduction of binding, implying that the binding sites might be retained but binding was affected, we think it might be due to the conformational change that could reduce the binding or accessibility of lipoproteins. Since this region locates closer to the membrane, it’s possible that it may change the interaction with the membrane. In the AF model, Δ353-415 region does not seem to be more hydrophobic than other regions (Fig. S2C).

(5) What was the point of having Figure 8? Showing the SCARF1 homodimers could form two types of dimers on the membrane surface proposed? The authors didn't have any data to support that.

Fig. 8 shows a potential model of the SCARF1 dimers on the cell surface by combining the structural information from crystals and AF predictions. The two dimers in the figure are identical but with different viewing angles. The lipoprotein binding sites are also indicated (Fig. 8).

**Recommendations for the authors:**

**Reviewer #1 (Recommendations For The Authors):**
The authors need to show examples of the electron density for both structures.

Electron density examples of the two structures are shown in Fig. S2A.

Figure (1)The figure does not show enough details of the structure. The text mentions hydrogen-bond and disulfide bonds that stabilize the loops, these should be shown.

Disulfide bonds of the two structures are shown in Fig. 1.

Figure (2)D) The full gel should be shown.E) Rather than just relying on changes in gel filtration elution volumes, the authors do the appropriate experiment and measure the hydrodynamic radius of the WT and mutant ectodomains by DLS. However, they need to show plots of the size distribution, not just mean radial values, in order to show if the sample is monodisperse.

The full gel and plots of DLS are shown in Fig. S3A-B.

Figure (3)I have concerns about the rigor of the experiments in panels A-D. The authors include a non-transfected control but do not appear to have treated non-transfected cells with the lipoproteins to evaluate the specificity of binding. Every cell binding assay (flow or confocal) must show the data from non-transfected cells treated with each lipoprotein, as each lipoprotein species could have a unique non-specific binding pattern. The authors show these controls in Figure 6, but these controls are necessary in every experiment.

In Fig. 3A, since several lipoproteins were included in the figure, we use non-transfected cells without lipoprotein treatment as a negative control. The OxLDL or AcLDL treated non-transfected cells were also used as negative controls and shown in Fig. 3B-C. LDL, HDL or OxHDL may have their own non-specific binding patterns, the treatment of LDL, HDL or OxHDL with the transfected cells all gave negative results (Fig. 3A and D).

Cell-surface of the SCARF1 variants is a major concern. The constructs the authors use are tagged with a GFP on the cytosolic side. However, the Methods to do indicate if they gate on GFP+ transfected cells for analytical flow. Such gating may have been used because the staining experiments in Figures 3 and 4 show uniform cell populations, whereas the staining done with an anti-SCARF1 Ab in S4 shows most of the cells not expressing the protein on the surface. Please clarify.

Data for the anti-SCARF1 Ab assay is gated for GFP in the revised Fig. S4, and the non-transfected cells are included as a control.

The authors must demonstrate cell-surface staining with an epitope tag on the extracellular side and clarify if the analyzed cells are gated for surface expression. The anti-SCARF antibody used in S4 may not recognize the truncated or mutant SCARFs equally. Cell-surface expression in the flow experiments cannot be inferred from confocal experiments because the flow experiments have a larger quantitative range.

Anti-SCARF1 antibody assay provides an estimation of the surface expression of the proteins. If the epitope of the antibody was mutated or removed in the mutants, most likely it would lose binding activity. Including an epitope tag on the ectodomain could be an option, but if truncation or mutation changes the conformation of the ectodomain, the accessibility of the epitope may also be affected, and addition of an extra sequence or domain, such as an epitope tag, may affect the surface expression of proteins sometimes.

In several places, the authors infer increased or decreased affinity from mean fluorescent intensity values of a single concentration point without doing appropriate dose-curves. These experiments need to be done or else the mentions of changes in apparent affinities should be removed.

We add a concentration for the WT interaction with OxLDL (Fig. S6, p.9) and the manuscript is also modified accordingly.

Figure 7The concentration of teichoic acid used to inhibit modified LDL binding should be indicated and a dose-curve analysis should be done comparing teichoic acid to some non-inhibitory bacterial polymer.

The concentration of teichoic acids used in the inhibition assays is 100 mg/ml (p.21). Unfortunately, we don’t have other bacterial polymers in the lab and not sure about the potential inhibitory effects.

**Reviewer #2 (Recommendations For The Authors):**
Major points:(1) The SCARF1 ECD contains three N-linked glycosylation sites (N289, N382, N393). It remains unclear whether these modifications are involved in SCARF1 binding to modified LDLs. Is it possible to design some experiments to investigate the effect of N-glycans on the recognition of modified LDLs? In particular, N382 and N393 are included in 353-415aa and the truncation mutant of SCARF1Δ353-415aa resulted in reduced binding with OxLDL in Fig.3G. Or whether the reduced binding is only due to the potential conformational changes caused by the deletion of the C-terminal region of the ECD?

A previous study regarding the N-glycans (N289, N382, N393) of SCARF1 (ref.17) has shown that they may affect the proteolytic resistance, ligand-binding affinity and subcellular localization of SCARF1, which is not quite surprising as lipoproteins are large particles, the N-glycans on the surface of SCARF1 could affect accessibility or affinity for lipoproteins. But the exact roles of each glycan could be difficult to clarify as they might also be involved in protein folding and trafficking.

The reduction of the binding of OxLDL for the mutant SCARF1 Δ353-415aa may be due to the conformational change or the loss of the glycans or both.

(2) The authors speculated that the dimeric form of SCARF1 may be more efficient in recognizing lipoproteins on the cell surface. Please highlight the critical region/sites for ligand binding in Figure 8 and discuss the structural basis of dimerization improving the binding.

The binding sites for lipoproteins on SCARF1 are indicated in Fig. 8. According to our data, it might be possible the conformation of the dimeric form of SCARF1 makes it more accessible to the ligands on the cell surface as implied by flow cytometry (p.14-15), but still needs further evidence on this.

(3) Could the two salt bridges (D61-K71, R76-D98) observed in f1 crystals be found in f2 crystals? They seemed to be a little far from the defined dimeric interface (F82, S88, Y94) and how important are these to SCARF1 dimerization?

The two salt bridges observed in f1 crystal are not found in f2 crystal (distances are larger than 5.0 Å), suggesting they are not required for dimerization (p. 7-8), but may be helpful in some cases.

(4) The monomeric mutants (S88A/Y94A, F82A/S88A/Y94A) exhibited opposite affinity trends to OxLDL in ELISA and flow cytometry. The authors proposed steric hinderance of the dimers coated onto the plates as the potential explanation for this observation. However, the method of ELISA stated that OxLDLs, instead of SCARF1 ECD, were coated onto the plates. So what's the underlying reason for the inconsistency in different assays?

Thanks. ELISA was done by coating OxLDLs on the plates as described in the Methods. But still, a dimeric form of SCARF1 may only bind one OxLDL coated on the plates due to steric hinderance. We correct this on p.12.

Minor points:(1) Figure 2D and Figure S3 - please label the molecular weight marker on the SEC traces to indicate the native size of various purified proteins.

The elution volume of SEC not only reflects the molecular weight, but it’s also affected by the conformation or shape of protein. The ectodomain of SCARF1 has a long curved conformation, the elution volumes of the monomeric or dimeric forms of SCARF1 do not align well with the standard molecular weight marker and elute much earlier in SEC. We include the standard molecular weight marker in Fig. S3C-D.

(2) Could the authors provide SEC profiles of f1 and f2 that were used in crystallographic study?

The SEC profiles of f1 and f2 for crystallization are shown in Fig. S5 (p.6).

(3) The legend of Figure 3A states that the NC in flow cytometry assay represents the non-transfected cells, but please confirm whether the NC in Fig. 3A-C corresponds to non-transfected cells or no lipoprotein.

NC in Fig. 3A represents the non-transfected cells, and no lipoproteins were added in this case as several lipoproteins are included in Fig. 3A. The lipoprotein (OxLDL or AcLDL) treated non-transfected cells (NC) were shown in Fig. 3B-C as negative controls.